# Sex and Limb Differences in Lower Extremity Alignment and Kinematics during Drop Vertical Jumps

**DOI:** 10.3390/ijerph18073748

**Published:** 2021-04-03

**Authors:** Youngmin Chun, Joshua P. Bailey, Jinah Kim, Sung-Cheol Lee, Sae Yong Lee

**Affiliations:** 1Department of Movement Sciences, University of Idaho, Moscow, ID 83843, USA; ychun@uidaho.edu (Y.C.); joshuabailey@uidaho.edu (J.P.B.); 2Department of Physical Education, Yonsei University, Seoul 03722, Korea; jjiinah.kim@gmail.com (J.K.); cheol3192@yonsei.ac.kr (S.-C.L.); 3Yonsei Institute of Sport Sciences and Exercise Medicine, Yonsei University, Seoul 03722, Korea

**Keywords:** posture, landing mechanics, joint kinematics, biomechanics

## Abstract

Sex and limb differences in lower extremity alignments (LEAs) and dynamic lower extremity kinematics (LEKs) during a drop vertical jump were investigated in participants of Korean ethnicity. One hundred healthy males and females participated in a drop vertical jump, and LEAs and LEKs were determined in dominant and non-dominant limbs. A 2-by-2 mixed model MANOVA was performed to compare LEAs and joint kinematics between sexes and limbs (dominant vs. non-dominant). Compared with males, females possessed a significantly greater pelvic tilt, femoral anteversion, Q-angle, and reduced tibial torsion. Females landed on the ground with significantly increased knee extension and ankle plantarflexion with reduced hip abduction and knee adduction, relatively decreased peak hip adduction, knee internal rotation, and increased knee abduction and ankle eversion. The non-dominant limb showed significantly increased hip flexion, abduction, and external rotation; knee flexion and internal rotation; and ankle inversion at initial contact. Further, the non-dominant limb showed increased peak hip and knee flexion, relatively reduced peak hip adduction, and increased knee abduction and internal rotation. It could be suggested that LEAs and LEKs observed in females and non-dominant limbs might contribute to a greater risk of anterior cruciate ligament injuries.

## 1. Introduction

According to the National Collegiate Athletic Association, non-contact anterior cruciate ligament (ACL) injuries are one of the most common sports-related injuries, whose rates have steadily increased by 1.3% each year [1]. Females face an increased risk of non-contact injury [2], with rates 2 to 5 times higher than those of males [3,4,5,6]. Others have investigated the possible contributing factors for the discrepancy in non-contact ACL injury rates between sexes, such as lower extremity alignment (LEA) and kinematics (LEKs) during sports-related movements [7,8,9,10,11,12,13,14,15,16,17], but the findings are contradictory.

The LEA variables, such as pelvic tilt, Q-angle, genu recurvatum, and navicular drop, are potential risk factors for non-contact ACL injury [7,8] that show sex differences. Females demonstrate greater anterior pelvic tilt, internally rotated hip, and knee valgus with hyperextension alignments than males [7,8]. These are considered to affect other LEAs and lead to kinematics that increase risk for non-contact ACL injury. Specifically, increased anterior pelvic tilt leads to knee hyperextension in standing posture because the internal knee extensor moment counteracts the increased external hip flexion moment by flexed hip position [18]. Individuals with internally or externally rotated hip and knee valgus alignments revealed greater average knee abduction angles across the entire landing phase of the drop vertical jump task than those with neutral LEAs [9]. However, the peak knee abduction angle between individuals with greater and lower Q-angle during running [19] and single-leg squat [20] was similar.

Previous studies [10,11,12,13,14,15,16] have attempted to illustrate the possible causes for females’ increased risk of non-contact ACL injury by sex differences in LEKs. Females reported greater ankle plantarflexion at initial contact (IC) [11,14,16,17], reduced peak dorsiflexion [11,16], and peak foot pronation during the stance phase of landing activities [14]. Like excessive knee abduction followed by hip and knee internal or external rotation demonstrated in the actual ACL injury scenarios [21], sex differences have been reported at the knee joint, with females demonstrating reduced peak flexion [16,17], increased peak abduction [10,13,14] at IC [13], and peak internal rotation [10,14]. However, peak joint angles in the sagittal plane [11,12,13,15] and frontal plane at IC during landing activities [10,12,15] do not show sex differences in other studies.

The association of limb dominance to non-contact ACL injury rates has been examined in female soccer players who show the greater occurrence of non-contact ACL injury of their non-dominant limb (preferred support limb) than dominant limb (preferred kicking limb), whereas 70% of injuries occurred in dominant limbs of male soccer players [22]. It should be noted that the non-dominant limb exhibited reduced knee flexion with adduction, increased hip flexion, adduction, internal rotation at IC [23], and greater knee internal rotation [24] during cutting activities. These findings, though based on different activities, were contradictory. Further, based on the insufficient evidence available, it is difficult to conclude whether the non-dominant limb has a higher biomechanical risk of non-contact ACL injury than the contralateral limb.

Although LEAs and LEKs have been examined to identify the factors that contribute to greater non-contact ACL injury rates in females or limbs, there are discrepancies in the results as mentioned above. Since sex differences in LEAs and LEKs have not been simultaneously investigated in previous literature, it would be possible to have a limited understanding of how females’ and males’ LEA are associated with their LEKs. Therefore, the main purposes of this study were to investigate sex and limb differences (1) in LEAs related to increased non-contact ACL injury risk and (2) in LEKs during a drop vertical jump. It was hypothesised that (1) females would have increased pelvic tilt, femoral anteversion, Q-Angle, and genu recurvatum compared with males; (2) females would have landing mechanics representative of non-contact ACL injury risk such as increased peak knee abduction, internal rotation, and ankle eversion angles with reduced peak knee flexion; (3) there would be no bilateral limb differences in LEAs; (4) the non-dominant limb would exhibit increased peak knee abduction, internal rotation, and ankle eversion angles with reduced peak knee flexion angle compared with the dominant limb.

## 2. Materials and Methods

### 2.1. Participants

The number of participants needed for this study was calculated by conducting a power analysis with 0.05 alpha level and power of 0.8 using the mean and standard deviation of anterior pelvic tilt [7]. The calculated number of participants was 49 for each group. A total of 100 healthy college students, non-athletes, (50 males: age = 22.4 ± 3.6 years, height = 1.76 ± 0.07 m, body mass = 72.7 ± 9.5 kg; 50 females: age = 22.1 ± 2.6 years, height = 1.63 ± 0.05 m, body mass = 56.4 ± 5.2 kg) were included in this study. They were free from lower extremity injuries over the past year and had no history of surgery. The current study was approved by the institutional review board for human subjects at the University. All participants completed a medical history questionnaire to determine eligibility and signed the informed consent upon arrival to the laboratory.

### 2.2. Anthropometric Measurements

Participants were asked to wear spandex shorts for measuring six selected LEAs—pelvic tilt, femoral anteversion, genu recurvatum, Q-angle, tibial torsion, and navicular drop—of both dominant and non-dominant limbs. The dominant limb was defined as the limb used by participants to kick a ball as far as possible [25]. All measurements were conducted by a single examiner (YC), and the intraclass correlation coefficients (ICC_2,3_) for each measurement were greater than 0.88 (Table 1).

Anterior pelvic tilt was measured using an inclinometer (PALM, Performance Attainment Associates, St. Paul, MN, USA) following the method described in previous studies [7,8]. Participants were asked to stand upright with toes facing forward and a shoulder-width stance. The anterior-superior iliac spine (ASIS) and the posterior-superior iliac spine were palpated, and each tip of the inclinometer’s arm was placed on ASIS and posterior-superior iliac spine to measure the angle with respect to the horizontal.

Craig’s test [26] was used to measure femoral anteversion with a bubble inclinometer (Baseline, Tech-Med Services Inc., Hauppauge, NY, USA), which was calibrated prior to each measurement to set zero degrees for the true vertical. Participants were instructed to lie chest down and flex their knee to 90 degrees. The inclinometer was placed on the proximal aspect of medial malleoli, and the examiner passively rotated the hip joint. The greater trochanter was palpated and tracked until it reached the most lateral position while passively rotating the hip joint. The angle determined by the inclinometer was recorded.

Q-angle was measured using a goniometer with the participant in the standing position with shoulder-width stance and toes facing forward [27]. Following palpation of ASIS, the axis of the goniometer was placed on the centre of the patella with the reference arm pointed to ASIS and the moving arm on the tibial tuberosity. The angle created by a line from ASIS to the centre of the patella and a line from the centre of the patella to tibial tuberosity was measured.

Genu recurvatum was measured in the supine position with a bolster under the distal tibia using a goniometer [8]. The tested leg was correctly repositioned to parallel the line from the medial to the lateral epicondyle of the femur with the treatment table by the examiner. The examiner palpated and marked the centre of the lateral malleoli and lateral aspect of the knee joint line using a pen. The reference arm was pointed to the greater trochanter, and the moving arm was pointed to the lateral malleoli. The axis of the goniometer was positioned on the lateral aspect of the knee joint line. The angle between the thigh and shank segments was measured.

Tibial torsion was measured in the prone position using a modified method [28]. The examiner positioned the participant’s thigh segment so that the imaginary line from medial to lateral epicondyle is parallel with the treatment table. The reference arm was placed on the posterior plantar surface of the foot and paralleled with the transmalleolar axis. The moving arm of the goniometer was aligned with the true vertical to measure tibial torsion.

The navicular drop was measured by subtracting navicular height in neutral standing position from its height in the standing subtalar joint neutral position [8]. The examiner first palpated and marked navicular tuberosity with participants in a bilateral standing position. The navicular height was measured using a height gage (Vernier Height Gage series 506, Mitutoyo, Kawasaki, Japan) in the neutral standing position. The examiner palpated both the medial and lateral aspects of the head of the talus. Participants were asked to invert and evert their ankle until the medial and lateral aspects of the head of the talus were equally palpated to measure the navicular height in the subtalar joint neutral position. 

### 2.3. Dynamic Measurements

Participants performed a 5-min self-selected warm-up prior to drop vertical jumps. Following the warm-up, participants were asked to put on the standardised laboratory shoes (FTY No. CLU 600001, Adidas, Herzogenaurach, Germany), and then 16 reflective markers (14 mm diameter) were attached on bony landmarks in accordance with the VICON Plug-in gait lower extremity model. Drop vertical jumps were performed on the 30 cm box placed at a horizontal distance, half of the participant’s height from two force platforms (ORG-6, AMTI, Watertown, MA, USA, Figure 1a). Participants were instructed to drop off from the box by leaning the body forward without a jump, land on the centre of each force platform, and immediately perform a maximal effort vertical jump. To magnify the effects of lower extremity alignment on the landing kinematics between males and females by minimizing arm movements, arms were controlled with the position of approximately 90 degrees shoulder abduction and elbow flexion with palms facing forward (Figure 1b). Participants were not instructed on how to land. Practice trials (minimum of 2) were provided so that participants were familiar with the task. A trial was eliminated if either foot landed off a force platform. The task was repeated until participants correctly performed three trials of drop vertical jump. 

Kinematic data during drop vertical jumps were collected using 3D motion capture system with 8 infrared cameras (VICON, Oxford Metric Ltd., Oxford, UK) at 200 Hz sampling rate. In addition, ground reaction force data were also obtained to determine the timing of IC and toe-off (TO) for each trial at 1000 Hz sampling rate and synchronised with kinematic data. Kinematic and kinetic data were low-pass filtered at 10 Hz with fourth-order Butterworth filter. The data were processed to obtain 3D lower extremity joint angles at IC and the peak joint angles using a customised LabView program (LabView 8.5, National Instrument, Austin, TX, USA). Trials were interpolated to 100% (101 data points) of the landing cycle. The landing cycle was defined as a period from IC to TO. and the mean and 95% confidence interval (CI) for each data point were calculated. 

### 2.4. Statistical Analysis

For the statistical analysis, 2-by-2 mixed model MANOVA was performed using SPSS software (IBM SPSS Statistics 24, IBM Inc., Chicago, IL, USA). The independent variables were sex (male and female) and limb (dominant and non-dominant limb). The dependent variables were LEAs and hip, knee, and ankle angles at IC and peak angles (hip, knee, and ankle) during the landing cycle in the three anatomical planes. The normality of each dependent variable was visually checked using Q-Q plot [29]. The alpha level was set at 0.05 for the statistical analysis. Also, the ensemble curve analysis with 95% CI was conducted to investigate sex and limb differences in the lower extremity joint angle patterns in the frontal and transverse plane.

## 3. Results

### 3.1. Anthropometric Measurements

There was no significant interaction effect between sex and limb (Wilks’ λ = 0.693, *F*_(24,75)_ = 1.832, *p* = 0.146, η^2^= 0.307), but significant sex main effect (Wilks’ λ = 0.306, *F*_(24,75)_ = 7.091, *p* < 0.001, η^2^ = 0.694) and limb main effect (Wilks’ λ = 0.386, *F*_(24,75)_ = 4.973, *p* < 0.001, η^2^ = 0.614) were observed in MANOVA results. In the univariate tests, females had greater pelvic tilt (*F*_(1,98)_ = 8.582, *p* = 0.004, η^2^ = 0.081), femoral anteversion (*F*_(1,98)_ = 3.985, *p* = 0.049, η^2^ = 0.039), and Q-angle (*F*_(1,98)_ = 69.531, *p* < 0.001, η^2^ = 0.415). Males possessed greater tibial torsion (*F*_(1,98)_ = 9.403, *p* = 0.003, η^2^ = 0.088) than females (Table 2). In addition, greater genu recurvatum was observed in the dominant limb compared with the non-dominant limb (*F*_(1,98)_ = 14.281, *p* < 0.001, η^2^ = 0.127). 

### 3.2. Dynamic Measurements

In the joint angles at IC, females exhibited decreased hip abduction (*F*_(1,98)_ = 21.667, *p* < 0.001, η^2^ = 0.181), decreased knee flexion (*F*_(1,98)_ = 10.767, *p* = 0.001, η^2^ = 0.099), decreased knee adduction (*F*_(1,98)_ = 23.803, *p* < 0.001, η^2^ = 0.195), increased ankle plantarflexion (*F*_(1,98)_ = 5.790, *p* = 0.018, η^2^ = 0.056), and increased ankle external rotation (*F*_(1,98)_ = 6.985, *p* = 0.010, η^2^ = 0.067) compared with males (Table 3). The non-dominant limb had increased hip flexion (*F*_(1,98)_ = 9.963, *p* = 0.002, η^2^ = 0.092), hip abduction (*F*_(1,98)_ = 26.044, *p* < 0.001, η^2^ = 0.210), hip external rotation (*F*_(1,98)_ = 6.273, *p* = 0.014, η^2^ = 0.060), knee flexion (*F*_(1,98)_ = 16.146, *p* < 0.001, η^2^ = 0.141), knee internal rotation (*F*_(1,98)_ = 15.581, *p* < 0.001, η^2^ = 0.137) and ankle inversion (*F*_(1,98)_ = 4.236, *p* = 0.042, η^2^ = 0.041) at IC compared with the dominant limb. 

In the peak joint angles, females showed increased peak hip adduction (*F*_(1,98)_ = 32.506, *p* < 0.001, η^2^ = 0.249), knee abduction (*F*_(1,98)_ = 29.462, *p* < 0.001, η^2^ = 0.231), ankle eversion (*F*_(1,98)_ = 4.155, *p* = 0.044, η^2^ = 0.041), and reduced peak knee internal rotation (*F*_(1,98)_ = 5.704, *p* = 0.019, η^2^ = 0.055) (Table 4). There were also significant main effects for limb in peak joint angles. The non-dominant limb experienced increased peak hip flexion (*F*_(1,98)_ = 18.293 *p* < 0.001, η^2^ = 0.157), knee flexion (*F*_(1,98)_ = 22.251, *p* < 0.001, η^2^ = 0.185), abduction (*F*_(1,98)_ = 12.278, *p* = 0.001, η^2^ = 0.111), and internal rotation (*F*_(1,98)_ = 27.780, *p* < 0.001, η^2^ = 0.221), and reduced peak hip adduction (*F*_(1,98)_ = 26.205, *p* < 0.001, η^2^ = 0.211) than the dominant limb.

Females showed different lower extremity joint angle patterns as compared with males regardless of limbs (Figure 2). Females showed reduced hip abduction and increased knee abduction through the entire landing cycle. Also, females exhibited reduced ankle inversion from 0% to 8%, 14% to 18%, and 98% to 100% of the landing cycle. In the transverse plane, it was observed that females revealed reduced hip internal rotation (19% to 83% of cycle), knee internal rotation (4% to 100% of cycle), and ankle external rotation at the beginning of the landing cycle (0% to 1%). Further, there were differences in joint angle patterns between limbs regardless of sex (Figure 3). The non-dominant limb displayed increased hip abduction (0% to 100% of cycle), knee abduction (10% to 89% of cycle), hip external rotation (0% to 38% and 56% to 100% of cycle), knee internal rotation (0% to 100% of cycle), reduced ankle external rotation (4% to 7% of cycle), and increased ankle internal rotation (97% to 100% of cycle). There was no bilateral difference in ankle frontal plane angle pattern.

## 4. Discussion

The major findings of this study were as follows: (1) sex differences in LEAs were observed in pelvic tilt, femoral anteversion, Q-angle, and tibial torsion; (2) sex differences in LEKs were observed in hip, knee, and ankle frontal, knee and ankle sagittal, and knee and ankle transverse plane angle during the drop vertical jump; (3) genu recurvatum revealed bilateral difference; (4) differences between dominant and non-dominant limbs in LEKs were identified in hip and knee angles in all three anatomical planes, and ankle frontal angle.

Sex differences in LEAs were mostly in line with those reported previously [7,8], but females in this study didn’t possess greater genu recurvatum than males as opposed to previous studies (Figure 4). Females possessed more anteriorly tilted pelvis, internally rotated femur, and knee valgus alignment than males. The anterior pelvic tilt value in the present study was close to not only healthy adults’ normal range (9 to 12 degrees) regardless of sexes and measuring methods [30,31,32,33], but also previously reported mean values (9 degrees for males and 11 degrees for females) [7,8,34]. The greater femoral anteversion results obtained in this study in females were analogous to the findings in previous studies [7,8,35]. Magee [26] showed that the normal range for femoral anteversion measured by Craig’s test was from 8 to 15 degrees in both males and females. It has been previously reported that females had an increase of approximately 3 degrees in femoral anteversion than males [7,8,35]. Average Q-angle ranges between 5 and 15 degrees for males and 10 and 19 degrees for females [7,8,27,36,37,38,39,40], and a Q-angle of more than 15 degrees for males and 20 degrees for females are considered excessive [41]. Regardless of limb dominance, Q-angles in males (14.4 degrees) and females (21.6 degrees) showed that the females in our study fell within the excessive Q-angle range, whereas the males were on the high end of the average. It remains unknown whether the average values obtained from previous studies apply to the population presented in the current study since all participants were Korean; it is presumed that participants in previous studies were of mixed ethnicities as most studies were conducted in the US. Omololu et al. [42] reported a disagreement between Q-angle averages in Nigerian and previously reported values. They concluded the different Q-angle might be attributed to anatomical differences such as pelvic width and femoral length between races. 

Although our SEM of tibial torsion was slightly greater than a previous study [8], the tibial torsion of males (approximately 20 degrees) was similar to that previously reported [8,43]. However, a significant sex difference in tibial torsion was observed in this study (approximately 17 degrees) due to lesser tibial torsion in females as compared to the previous study (approximately 19 degrees) [8]. Tamari et al. [44] reported males had tibial torsion 9 degrees greater than that in females, regardless of ethnicity and age, using a different method of measurement. It was also reported that the tibial torsion measured by computed tomography was about 38 degrees in normal adults regardless of sex [45], which are nearly 20 degrees more than that obtained by goniometry, in the current study, and two previous reports [8,43]. This disagreement between measuring methods may be due to the measurement of the rotational angle of the distal part of the tibia in goniometry, whereas the computed tomography and two digital inclinometers measure the summation of rotational angles on both proximal and distal parts of the tibia.

As expected, females displayed more erect and toe-out landing posture with a relatively medial collapsed limb, which may contribute to the increased risk of non-contact ACL injury. The increased ankle plantarflexion [11,14,16,17] and reduced knee flexion [16,17] at IC observed in healthy females suggests that this joint positioning difference in females represents a shift in females to an ankle-landing strategy [17]. This posture, especially reduced knee flexion at IC, is believed to be vulnerable to non-contact ACL injuries created by tibial anterior shear force generation by quadriceps contraction at the shallow knee flexion angle [21,46]. It was also notable that the females in our study exhibited increased relative peak hip adduction, knee abduction, and ankle eversion during the landing cycle. However, Kernozek et al. [10] reported that females had increased peak knee abduction and ankle pronation during bilateral drop landing, compared with males, but not in peak hip frontal angle. The disagreement between studies in sex difference in peak hip frontal angle may be attributed to the relatively small sample size in the previous study (15 males and 15 females), compared with our study. Interestingly, males showed an increase in peak knee internal rotation, which is considered a major risk factor for non-contact ACL injury [21] and was observed in females during drop vertical jump [14] and single-leg drop landing [47]. A possible explanation may be that reduced peak knee internal rotation is attributed to increased hip internally rotated alignment (i.e., increased femoral anteversion) in females as postural compensation to an externally rotated tibia [48]. It has been shown that individuals with internally rotated hips and valgus knee alignment, as opposed to those with externally rotated hips and valgus knee alignment, demonstrate increased external knee external rotation moment [9].

Since non-dominant limbs possessed reduced genu recurvatum compared with dominant limbs in both males and females, the dominant limb may have greater joint laxity than non-dominant limb. Genu recurvatum was measured in a relaxed supine position with a bolster under the distal shank, which supports the knee joint by only passive structures, such as a ligament and joint capsule. Lin et al. [49] measured and demonstrated that dominant limbs displayed greater knee joint laxity than non-dominant limbs in young females. In addition, it was reported that genu recurvatum was associated with anterior knee joint laxity and a strong predictor of the laxity regardless of sexes [50]. Thus, the greater genu recurvatum in the dominant limb versus the contralateral limb may be the result of increased knee joint laxity, enhancing the risk of non-contact ACL injury.

We also observed bilateral differences in dynamic LEKs. Non-dominant limbs revealed greater hip and knee flexion, abducted hip and inverted ankle, and externally rotated hip and internally rotated knee at IC, increased peak hip and knee flexion, relatively reduced peak adduction, and increased knee abduction and internal rotation compared with dominant limb. Most of our findings regarding bilateral differences were consistent with those observed in a previous study [23], despite different landing tasks. It is speculated that both males and females, despite differences in their landing strategies, prefer to shift the centre of mass to the dominant limb before landing. This shift is demonstrated by greater muscle activation in the dominant limb prior to landing [51]. Also, Ball et al. [52] showed greater mean and peak vertical ground reaction force, as well as earlier and longer contact with the ground in the dominant limb. Even if the centre of mass shifted toward the dominant limb, the limb showed well-controlled movement patterns. The non-dominant limb, meanwhile, revealed more oscillated movement patterns in both frontal and transverse planes with the increased knee abduction and internal rotation (Figure 3). This finding may suggest that lower extremity muscles on the non-dominant limb are not activated as much as the contralateral limb, which increases the likelihood of non-contact ACL injury.

There were several limitations to this study. First, the drop vertical jump was performed in the arms-controlled posture. Since actual sports-related movements require arm swings to jump as high as possible for vertical jump following landing, the controlled task in this study might reduce the variability of movement patterns. Secondly, the motion capture system with skin-mounted markers was used for this study. The use of skin-mounted markers may cause substantial errors in knee motions in both frontal and transverse planes than sagittal plane [53]. Additionally, the box height was limited to 30 cm for both males and females. This condition might affect the sex differences in landing mechanics due to the discrepancy of averaged statures and body mass between males and females.

## 5. Conclusions

This study demonstrated sex and limb dominance differences in LEAs and LEKs during drop vertical jumps in participants of Korean ethnicity. According to the LEAs and LEKs measured, females and the non-dominant limb possess greater risk factors of non-contact ACL injuries. Specifically, females possess more anteriorly tilted pelvis, internally rotated femurs, and valgus knee posture, and non-dominant limb had hyperextended knee as compared to the dominant limb. Also, females showed more erected and medially collapsed position during drop vertical jumps. The non-dominant limb displayed greater changes in joint angles in frontal and transverse plane than dominant limb during drop vertical jumps. The findings of the current study reinforce sex and limb differences related to non-contact ACL injuries reported previously and present the possibility of interracial differences in LEAs and LEKs. Thus, in future studies, it would be necessary to develop and examine a training protocol reducing medial collapse of the knee for females and angular displacement in the frontal and transverse planes for non-dominant limbs.

## Figures and Tables

**Figure 1 ijerph-18-03748-f001:**
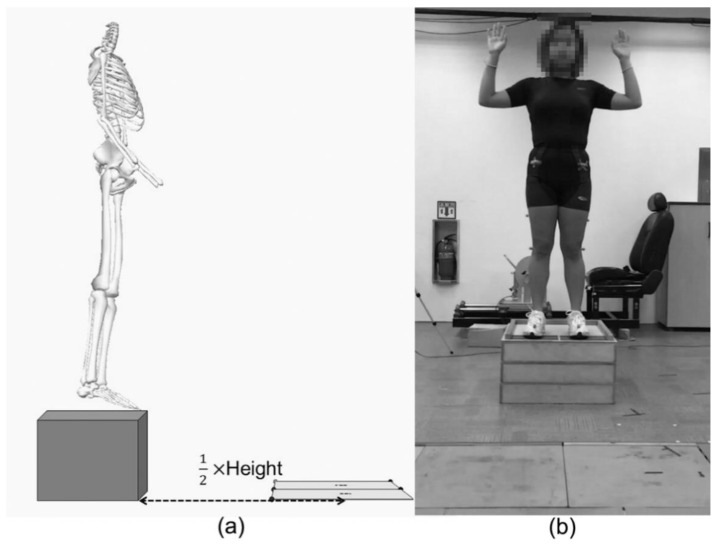
(**a**) The box was located to half of each participant’s height from the centre of force platforms. (**b**) Participant’s posture for drop vertical jump. Arms were controlled with approximately 90 degrees abduction and elbow flexion.

**Figure 2 ijerph-18-03748-f002:**
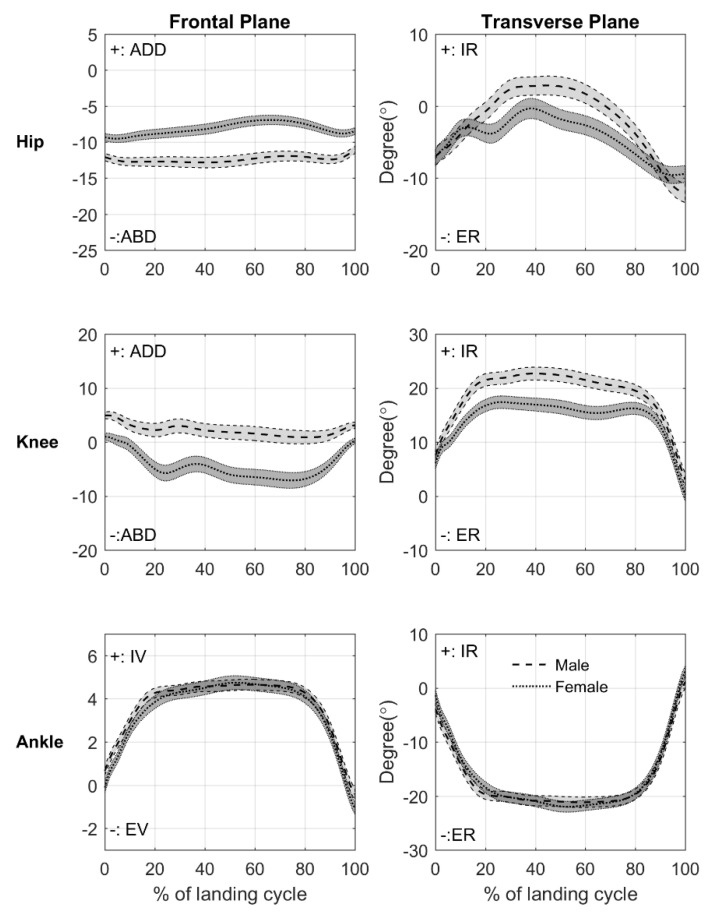
Sex differences in the hip and knee joint angle patterns in frontal and transverse planes from initial contact to toe-off (0–100% of the landing cycle) during the stance phase of drop vertical jump. The dashed line with light grey area represents the mean ± 95% CI for males, and the dotted line with dark gray area indicates the mean ± 95% CI for females. The positive values present hip and knee adduction (ADD), ankle inversion (IV), and hip, knee, and ankle internal rotation (IR), and the negative values present hip and knee abduction (ABD), ankle eversion (EV), and hip, knee, and ankle external rotation (ER).

**Figure 3 ijerph-18-03748-f003:**
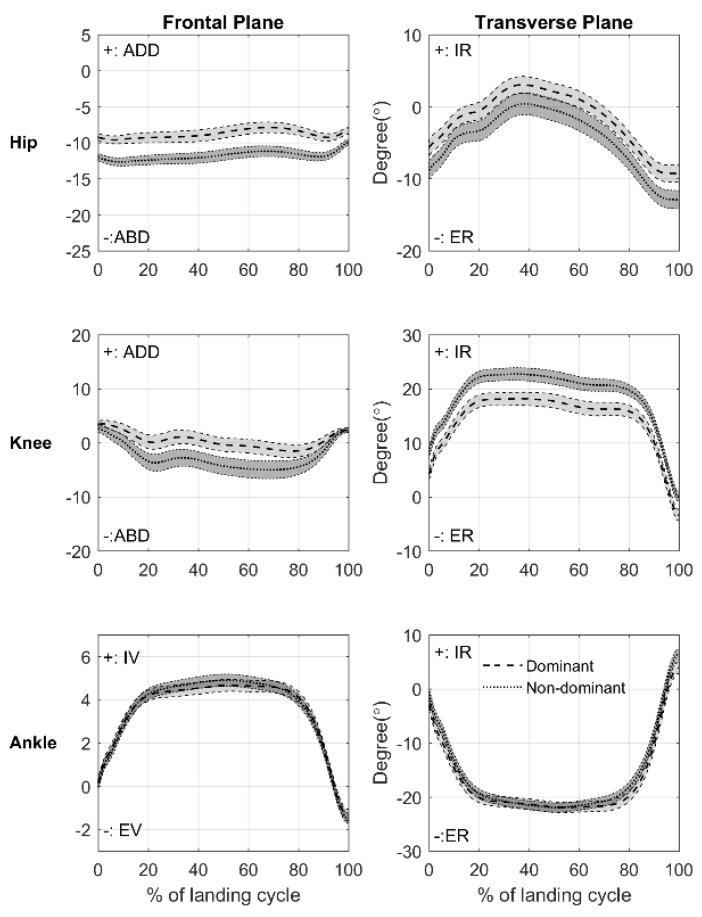
Bilateral differences in the hip and knee joint angle patterns in frontal and transverse planes from initial contact to toe-off (0–100% of landing cycle) during the stance phase of drop vertical jump. The dashed line with light grey area represents the mean ± 95% CI for dominant limb, and the dotted line with dark gray area indicates the mean ± 95% CI for non-dominant limb. The positive values present hip and knee adduction (ADD), ankle inversion (IV), and hip, knee, and ankle internal rotation (IR). The positive values present hip and knee adduction (ADD), ankle inversion (IV), and hip, knee, and ankle internal rotation (IR), and the negative values present hip and knee abduction (ABD), ankle eversion (EV), and hip, knee, and ankle external rotation (ER).

**Figure 4 ijerph-18-03748-f004:**
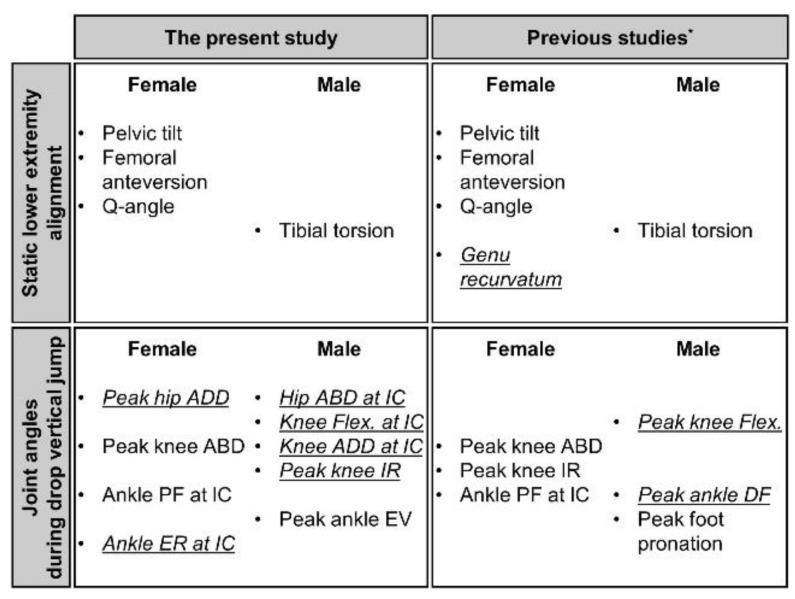
The comparisons of major findings of this study with previously reported differences in lower extremity alignments (LEAs) and lower extremity kinematics (LEKs) during landing activities between males and females. The *italic and underlined variables* are the disagreement between the findings of the present study and previous studies. * indicates references [7,8,9,11,12,15,17,27,35,36,37,38,39,40,42,43,44,45] were cited. *Note.* ABD: abduction; ADD: adduction; Flex: Flexion; IR: internal rotation; PF: plantarflexion; DF: dorsiflexion; EV: eversion; ER: external rotation; IC: initial contact.

**Table 1 ijerph-18-03748-t001:** Intraclass correlation coefficient (ICC) and standard error of the measurement (SEM) for each lower extremity alignment measurement.

Measurement	ICC	SEM
Pelvic tilt	0.88	1.35
Femoral anteversion	0.94	0.91
Q-angle	0.93	0.94
Genu recurvatum	0.90	1.09
Tibial torsion	0.93	1.97
Navicular drop	0.95	0.53

**Table 2 ijerph-18-03748-t002:** Lower extremity alignment measurements for sex and limb.

	Male		Female	
	Dominant	Non-Dominant	Dominant	Non-Dominant
Pelvic tilt *	8.6 (3.3)	9.0 (3.9)	10.8 (3.5)	10.8 (3.7)
Femoral anteversion *	7.1 (6.9)	7.7 (6.1)	10.1 (6.3)	9.6 (6.7)
Q-angle *	14.6 (4.7)	14.1 (4.7)	21.8 (5.4)	21.4 (4.7)
Genu recurvatum ^†^	2.1 (4.1)	1.3 (3.9)	3.1 (2.9)	2.3 (3.3)
Tibial torsion *	20.0 (7.4)	20.1 (6.3)	16.6 (6.2)	17.2 (5.7)
Navicular drop	7.9 (3.8)	8.4 (4.0)	7.5 (3.8)	8.0 (3.4)

Note. All variables were measured in degrees. Mean (SD); * indicates significant main effect for sex (*p* < 0.05). ^†^ indicates significant main effect for limb (*p* < 0.05).

**Table 3 ijerph-18-03748-t003:** Lower extremity angles at initial contact.

	Male		Female	
	Dominant	Non-Dominant	Dominant	Non-Dominant
Hip flexion/extension ^†^	36.3 (7.3)	37.0 (8.5)	37.1 (5.8)	38.7 (5.2)
Hip adduction/abduction *^†^	−10.8 (3.9)	−13.0 (4.1)	−7.6 (3.7)	−10.9 (3.9)
Hip internal/external rotation ^†^	−6.2 (9.8)	−8.9 (11.4)	−6.1 (10.4)	−9.8 (10.2)
Knee flexion/extension *^†^	25.3 (7.1)	27.1 (7.6)	20.7 (5.9)	23.9 (6.0)
Knee adduction /abduction *	4.9 (4.6)	4.6 (4.6)	1.7 (4.3)	0.5 (5.1)
Knee internal/external rotation ^†^	4.4 (9.4)	6.5 (9.0)	1.1 (8.1)	7.1 (8.5)
Ankle dorsiflexion/plantarflexion *	−10.6 (11.6)	−11.0 (10.6)	−16.8 (10.9)	−14.8 (11.3)
Ankle inversion/eversion ^†^	−3.5 (8.7)	−0.1 (8.9)	0.6 (8.4)	1.3 (8.6)
Ankle internal/external rotation *	0.3 (1.9)	0.1 (2.0)	−0.7 (2.0)	−0.6 (1.8)

Note. All variables were measured in degrees. The first of the two variables listed for each joint angle corresponds to a positive value. Mean (SD); * indicates significant main effect for sex (*p* < 0.05). ^†^ indicates significant main effect for limb (*p* < 0.05).

**Table 4 ijerph-18-03748-t004:** Peak lower extremity angles during a cycle.

	Male		Female	
	Dominant	Non-Dominant	Dominant	Non-Dominant
Hip flexion ^†^	75.5 (15.1)	79.9 (12.1)	78.6 (15.3)	81.6 (11.6)
Hip adduction *^,†,§^	−6.8 (4.7)	−9.7 (3.9)	−3.1 (4.1)	−5.9 (4.3)
Hip internal rotation	6.1 (10.2)	4.9 (12.5)	5.9 (9.6)	2.5 (10.5)
Knee flexion ^†^	100.4 (15.7)	101.7 (15.7)	97.8 (12.9)	100.3 (14.7)
Knee abduction *^,†,§^	−2.3 (7.9)	−5.4 (9.0)	−9.8 (9.0)	−14.9 (11.6)
Knee internal rotation *^,†^	23.9 (9.9)	27.4 (9.6)	18.7 (9.3)	24.4 (9.6)
Ankle dorsiflexion ^‡^	34.2 (5.0)	33.7 (5.0)	34.3 (4.5)	35.7 (6.1)
Ankle eversion *^,§^	−1.1 (1.9)	−1.5 (1.8)	−2.0 (2.0)	−1.9 (1.9)
Ankle external rotation ^‡,§^	−25.4 (7.9)	−23.0 (6.1)	−24.2 (7.7)	−26.0 (6.9)

Note. All variables were measured in degrees. Mean (SD); * indicates significant main effect for sex (*p* < 0.05). ^†^ indicates significant main effect for limb (*p* < 0.05). ^‡^ indicates significant interaction effect between sex and limb (*p* < 0.05). ^§^ indicates significant negative values indicate hip abduction, knee abduction, ankle eversion, and ankle external rotation.

## Data Availability

The data presented in this study are available on request from the corresponding author.

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
