# Peer review of "Sex and Limb Differences in Lower Extremity Alignment and Kinematics during Drop Vertical Jumps"

_ijerph, 2021, doi:10.3390/ijerph18073748_

Round 1
Reviewer 1 Report
All comments are in file.

Author Response
Dear Reviewer,
Thank you for your suggestions and comments on my manuscript. I have addressed all of your comments. The detailed responses have attached in a Word document format. Please find the attachment.

Reviewer 2 Report
This study gives a detailed biomechanical analysis between the sexes for the lower limb when landing from a height and has implication for explaining injury. It is generally well-written and use of a MANOVA to analyze the many outcome variables is a strong statistical approach.
Line 48: “Previous studies [11–17] has attempted” – change “has” to “have”
Line 53: Change “Sex” to “sex”
Lines 131-132: “The examiner realigned the thigh segment to parallel the line from the medial to lateral epicondyles of the femur with the treatment table.” The wording in this sentence is confusing. Please consider re-wording.
Legend for Figure 4: “This is a figure. Schemes follow the same formatting” – should this be part of the figure? I suggest deleting.
Discussion, line 249: I don’t think it is necessary to re-state the purpose of the study here. I suggest deleting this first sentence.
Lines 250-254: The word “the” can be deleted from the start of each of these numbered points when summarizing the important results.
Lines 257-258: “but the partially disagreed results were also observed” – some re-wording is required here.
Figure legends should not contain unexplained abbreviations (i.e. the reader should not have to refer to the text for the abbreviation explanation, as figures should “stand alone” from the text). For figures 3 and 4, please define the abbreviations IC and TO in the figure legend.
Figure 5 legend: Define the abbreviations LEAs and LEKs in the legend
Practical application of the results: According to your results can you give any recommendation on how to reduce knee injuries? Can you give recommendations on how to alter kinematics when landing or how to train the athlete to prevent injury?
Author Response

(The authors gave the same response as above.)

Reviewer 3 Report
Well done to the authors of this study, for it clear design, method, and balanced discussion and conclusions.
I will make brief points that would be helpful:
- given the M:F differences, please further comment about basic load/vectors, given the overt differences in height and weight between sexes
- what were the warm ups of choice? Any differences between sexes?
- Is kick a ball the best test of limb dominance for participants who may never kick a ball?
- Please include discussion re SEMs, eg tibial torsion difference of 3 degrees between M:F - but SEM almost 2 degrees
- standardised arm position versus natural jump - please comment
Otherwise, well done, and a nice study within Korean ethnicity - which of course also limits external validity, but you have well stated this point.
Author Response

(The authors gave the same response as above.)

Round 2
Reviewer 1 Report
The Authors responded to all comments and revised the manuscript. I have no further comments and I can recommend the paper for publication in present form.